# Cross-Sectional Survey on BNT162b2 mRNA COVID-19 Vaccine Serious Adverse Events in Children 5 to 11 Years of Age: A Monocentric Experience

**DOI:** 10.3390/vaccines10081224

**Published:** 2022-07-30

**Authors:** Silvia Bloise, Alessia Marcellino, Beatrice Frasacco, Pietro Gizzone, Claudia Proietti Ciolli, Vanessa Martucci, Mariateresa Sanseviero, Emanuela Del Giudice, Flavia Ventriglia, Riccardo Lubrano

**Affiliations:** Dipartimento Materno Infantile e di Scienze Urologiche, Sapienza Università di Roma, UOC di Pediatria e Neonatologia-Polo Pontino, 04100 Latina, Italy; alessia.marcellino@uniroma1.it (A.M.); beatrice.frasacco@uniroma1.it (B.F.); pietro.gizzone@studenti.unicampania.it (P.G.); proietticiolli.1636558@studenti.uniroma1.it (C.P.C.); v.martucci@ausl.latina.it (V.M.); m.sanseviero@ausl.latina.it (M.S.); emanuela.delgiudice@uniroma1.it (E.D.G.); flavia.ventriglia@uniroma1.it (F.V.); riccardo.lubrano@uniroma1.it (R.L.)

**Keywords:** COVID-19, vaccine, children, adverse events

## Abstract

Objective: Our aim was to evaluate the safety of COVID-19 vaccine in children resident in the Latina Local Health Authority. Methods: We conducted a telephone survey among children aged 5–11 years receiving BNT162b2 mRNA COVID-19 vaccine between December 15 and 21. The main outcomes included the presence of allergic reactions or anaphylaxis, adverse events after 24–48 h, 7 and 20 days of taking the first and second doses of medications, and documented SARS-CoV-2 infection after vaccination. The information obtained was automatically linked to a spreadsheet and analyzed. Results: 569 children were enrolled. The mean age was 114 ± 4.24 months; there were 251 males in the study. The vaccine showed a favorable safety profile; no anaphylaxis or serious adverse events were reported. The most common symptoms both after the first and second dose were injection site reactions, asthenia, and headache. Injection site reactions were more frequent after the first dose (*p* = 0.01), while systemic symptoms were more frequent after the second dose (*p* = 0.022). These symptoms were more frequent in patients with comorbidities (*p* = 0.0159). Conclusion: Our findings confirm the safety of COVID-19 vaccine in children younger 11 years and could be useful to promote its diffusion in pediatric ages in order to achieve “herd immunity” and prevent the virus’s circulation.

## 1. Introduction

Two years after the onset of the pandemic, about 567 million cases of COVID-19 have already been reported worldwide and their increase in the last months has been an expression of the fifth wave of the SARS-CoV-2 [1,2,3,4].

The development and testing of vaccines against COVID-19 has been very rapid. In fact, already in the first half of 2021 many vaccines with different mechanisms of action were introduced in many countries. The efficacy of these vaccines has been demonstrated through phase 3 clinical trials [5,6,7,8,9,10,11,12]. These data are important, but to establish the true effectiveness of a vaccine it is necessary to assess the impact on the general population (that is, the reduction in the risk of infection and disease among vaccinated people) [13,14].

Since the introduction of the vaccine campaign, many reports have shown promising results with reductions in the incidence of infection, hospitalizations, and deaths in different countries [15,16,17,18,19,20,21,22,23,24]. Specifically, in Italy, according to data from the Istituto Superiore di Sanità, thanks to vaccination there have been more than 500,000 hospitalizations, more than 55,000 ICU admissions, and about 150,000 deaths avoided [25].

To date, the vaccination of the global population seems to be the only effective strategy to achieve the “herd immunity” and consequently prevent the virus’s circulation [26,27]. This requires the entire population to be vaccinated, including children and adolescents. In addition, since antibody response against SARS-CoV-2 has been shown to wane over time, more booster doses are needed [28,29,30,31].

Until December 2021, vaccination has been authorized for COVID-19 prevention from the age of 11 years [32,33]. On 7 December 2021 the authorization was expanded to include children aged from 5 to 11 years [34,35,36,37]. Vaccinating this second age group could help, in association with the implementation of preventive measures in children [38,39,40,41], to mitigate downstream effects of the pandemic, such as social isolation and disruption of education, and allow for a return to normal life. However, the issue of vaccination in the pediatric age group has been very debated and there is still a lot of fear and perplexity among parents and pediatricians. Therefore, it is important to assess carefully the risks and benefits of the vaccination in children [42] and in order to justify childhood vaccination, it is necessary an effort by the scientific community to report all experiences and possible adverse events related to the vaccine. This could improve community knowledge about this topic, debunking numerous fake news and increasing the compliance and consent of caregivers. In this context, we have conducted a cross-sectional survey at our center on the effects of BNT162b2 mRNA COVID-19 vaccine administered to children 5 to 11 years old. The primary aim was to evaluate the safety of the vaccine through the assessment of adverse events reported by parents 24-48 h, 7 days, and 20 days after the first and the second dose. The secondary aim was to evaluate the difference between first and second dose in terms of tolerability. The tertiary aim was to evaluate the difference in incidence of SARS-CoV-2 infection with an onset of seven days after first and second dose between the vaccinated and unvaccinated pediatric population residing in the Local Health Authority (ASL) of Latina.

## 2. Materials and Methods

This was a prospective, cross-sectional study conducted by the Pediatric Unit of Santa Maria Goretti Hospital, in Latina—Sapienza University of Rome (Polo Pontino) from 15 December 2021 to 31 January 2022. The Institutional Review Board of the Maternal and Child Health Department of the Local Health Authority of Latina approved the study protocol. This report follows the STROBE reporting guideline for cross-sectional studies.

We included all children aged 5–11 years undergoing first dose of vaccination with BNT162b2 mRNA COVID-19 vaccine between December 15 and 21. The exclusion criterion was previous SARS-CoV-2 infection within three months prior to the date of initiation of vaccination and an age greater than 11 years old or less than 5 years old. The vaccination protocol included two doses of vaccine spaced 21 days apart. In addition, only a single dose of vaccine was provided for children who had been previously infected for less than 12 months [43]

At the time of vaccination, parents gave written consent to receive the vaccine and were informed that they would be contacted by a doctor in our department to check for any side effects after vaccination.

We developed a survey with a web-form link to collect data (Google forms). The survey was conducted through a questionnaire administered, after a verbal informed consent, by telephone by the doctors of the department of pediatrics of Santa Maria Goretti Hospital to parents of children 5–11-years-old undergoing vaccination with BNT162b2 mRNA COVID-19 vaccine between December 15 and 21. 

The questionnaire included specific questions: children’s demographic characteristics, comorbidities, previous SARS-CoV-2 infection and clinical course, presence of allergic reactions or anaphylaxis after first and second doses, presence of adverse events after 24–48 h, 7 days and 20 days, after the first and second doses and any medications taken for these symptoms, and documented SARS-CoV-2 infection after vaccination.

The information obtained was automatically linked to a spreadsheet and analyzed.

To evaluate the difference in incidence of SARS-CoV-2 infection between the vaccinated and unvaccinated children of the ASL of Latina, we considered two time periods: the first from seven days after the first dose (December 22) to January 4; and the second from seven days after the second dose (January 12) to January 31. Then, we compared the number of children infected in the unvaccinated group and in the vaccinated group in the two time periods of study.

### Statistical Analysis

The statistical analysis was performed with JMP^®^ 16.1.0 program for Mac (SAS Institute Inc., Rome, Italy). The qualitative variables were described as the distribution of absolute frequencies and percentages. We used Fisher’s test to compare categorical variables and Chi square with Yates’ correction. A *p* value < 0.05 was considered significant.

## 3. Results

### 3.1. Clinical and Demographic Characteristics of Patients

We enrolled 569 children 5-11 years old receiving the BNT162b2 vaccine between 15 December 2021 to 11 January 2022. The mean age was 114 ± 4.24 months; 251 of these individuals were male.

Eighty-point-seven-percent of patients had no comorbidity, while nineteen-point three-percent had comorbidity (46% allergies, 24% asthma, 9% gastrointestinal diseases, 8% heart diseases, 7% neurological diseases 3% nephrological diseases, 2% rheumatological diseases, 1% other diseases).

Ninety-three-point-three percent of patients had not experienced SARS-CoV-2 infection at the time of vaccination, while six-point-four percent had a positive history of previous SARS-CoV-2 infection. Of these last few, forty-one-point-seven percent of children were asymptomatic, while fifty-eight-point-three percent showed mild symptoms. The most common symptoms of previous infection were: fever (71.4%), upper respiratory symptoms (42.8%), headache (33.4 %), gastrointestinal symptoms (14.2%), asthenia (9.5%), and arthralgias (4.8%).

The clinical and demographic characteristics are summarized in Table 1.

### 3.2. Adverse Events 24–48 h after First Dose

24–48 h after the first dose, sixty point one of participants reported local and/or systemic events and thirty-nine- point nine percent had no symptoms. The most common symptoms reported were: injection site reactions (95.6%), mainly pain (90.4%), redness (2.3%), swelling (2.9%) and induration of the site (0.2%). Systemic adverse effects were: asthenia (10.8%), headache (8.1%), gastrointestinal symptoms (4%), fever (2.9%), arthromyalgia (2.6%), lymphadenopathy (1.4%), irritability (1.7%), drowsiness (1.4%), epistaxis (0.29%), and paresthesias of the limbs (0.3%).

No thromboses or hypersensitivity adverse events or vaccine-related anaphylaxis were seen.

Detailed graphical representation of symptoms experienced after first dose of BNT162b2 mRNA COVID-19 vaccine is shown in Figure 1.

Local and systemic events were mild and typically resolved within a few days (12–24 h: 42.2 %; 12 h: 30.2 %; 24–48 h: 20.2 %; 48–72 h: 7.3 %).

Only twenty-four -point four percent of children took medications for these symptoms. The most commonly used medications were: acetaminophen (84.1 %), non-steroidal anti-inflammatory drugs (7.3%), and local topical creams (9.7%).

### 3.3. Adverse Events 7 Days after First Dose

Only one point nine percent of children experienced symptoms seven days after the first dose, while ninety-eight-point-one percent showed no symptoms. The symptoms reported were: persistence of pain at the injection site (50%), headache (20%), fever (20%), and asthenia (10%). The duration of these symptoms was: 12–24 h: 66.7%; 12 h: 11.1%; 48–72 h: 22.2%.

Thirty-three-point-three percent of children took medications for these symptoms. The most commonly used medications were: acetaminophen (66.7%), non-steroidal anti-inflammatory drugs (33.3%), and local topical creams (33.3%).

Seven days after first dose, there were twenty-seven (5.4%) cases of COVID-19 infection confirmed by molecular swab.

The infection was symptomatic in sixty -point seven percent of cases and asymptomatic in thirty-nine- point three percent of cases. The symptoms reported were mild and were fever (52.9%), headache (47%), upper respiratory symptoms (41.1%), gastrointestinal symptoms (11.7%), asthenia (11.7%), and arthralgias (11.7%).

### 3.4. Adverse Events 20 Days after First Dose

Ninety-nine-point-six percent of children not experienced symptoms 20 days after the first dose; Only two children (0.4%) experienced symptoms 20 days after the first dose. The symptoms were inappetence and intercostal pain. These symptoms were mild with spontaneous resolution in 48–72 h.

### 3.5. Adverse Events 24–48 h after Second Dose

Of the 569 children enrolled, 449 underwent the second dose and were surveyed, 49 did not undergo the second dose, and 71 did not answer the phone and did not complete the survey (Figure 2-flowchart).

The main reasons for not performing the second dose were: SARS-CoV-2 infection after the first dose (54.1%), quarantine status due to contact with positive subjects (25%), previous natural infection in the six months prior to vaccination (18.7%), fever from other causes (2%).

At 24–48 h after the second dose, fifty-one-point two percent of participants reported local and/or systemic events and forty-eight point eight had no symptoms. The most common symptoms reported were injection site reactions (95.8%), mainly pain (81.7%), redness (7.1%), swelling (6.2%) and induration of the site (1.25%). Systemic adverse effects were: asthenia (15.9%), headache (14.2%), fever (8.36%), gastrointestinal symptoms (6.27%), arthromyalgia (2.9%), linfoadenopatia (2.5%), shivers (1.6%), and irritability (0.8%),

No thromboses or hypersensitivity adverse events or vaccine-related anaphylaxis were seen after the second dose.

Detailed graphical representation of symptoms experienced after the second dose of BNT162b2 mRNA COVID-19 vaccine is shown in Figure 3.

Local and systemic events were mild and typically resolved within a few days (12–24 h: 43.8%; 12 h: 24.2%; 24–48 h: 22.5%; 48–72 h: 9.6%).

Only twenty-five-point-six percent of children took medications for these symptoms. the most commonly used medications were: acetaminophen (78.6%), non-steroidal anti-inflammatory drugs (11.4%), and local topical creams (11.4%).

No differences in reactogenicity were noted between participants who were SARS-CoV-2–positive at baseline and those who were SARS-CoV-2–negative at baseline (*p* = 0.86).

We found a difference between patients with comorbidities than healthy patients. In particular, the adverse events were more frequent in patients with comorbidities (*p* = 0.0159). Local and systemic events were always mild and the most common symptoms reported were the following: injection site reactions (53.7%) asthenia (5.5%), headache (3.7%), fever (1.8%), gastrointestinal symptoms (1.8%), irritability (1.8%),

### 3.6. Adverse Events 7 Days after Second Dose

Only zero-point-eight percent of children experienced symptoms seven days after the first dose, while ninety-nine-point-two percent showed no symptoms. The symptoms reported were: headache (50%), cough (25%) and lymphadenopathy (25%).

The duration of these symptoms was: 12–24 h: 25%; 24–48 h 25%; 48–72 h: 50%.

One-point-three percent of children took medications for these symptoms. The medication used was paracetamol acetaminophenin one hundred percent of cases.

Seven days after first dose, there were no cases of COVID-19 infection.

### 3.7. Adverse Events 20 Days after Second Dose

No children experienced symptoms 20 days after the second dose.

20 days after second dose, there were no cases of COVID-19 infection.

Difference in tolerability between first and second dose

We found some differences in reactogenicity between the two doses of BNT162b2 (*p* = 0.021).

In particular, injection site reactions were more frequent after the first dose (*p* = 0.01), while systemic symptoms such as fever was more frequent after the second dose (*p* = 0.022).

In contrast, there were no differences regarding other symptoms: Asthenia (*p*=0.27), headache (*p* = 0.11), arthromyalgia (*p* = 1), gastrointestinal symptoms (*p* = 0.45, lymphadenopathy (*p* = 1), drowsiness (*p* = 0.07), and irritability (*p* = 1).

### 3.8. Difference in Incidence of SARS-CoV-2 Infection between Unvaccinated and Vaccinated Children

In the ASL of Latina there are 37,003 resident children aged 5–11 years. In the period between December 22 and January 4, 538 of the unvaccinated children became infected, while only 27 of the 565 vaccinated children contracted the SARS-CoV-2 infection (*p* < 0.0001). In the period between January 12 and January 31, 3475 of the unvaccinated children became infected, while none of vaccinated children contracted the SARS-CoV-2 infection.

## 4. Discussion

### 4.1. Safety of COVID-19 Vaccine in Children 5–11 Years

Our results show that the vaccine BNT162b2 mRNA has a good safety profile in the 5–11-year-old age group. These findings are similar to the safety data reported from preauthorization trials for m-RNA vaccines COVID-19 in pediatric ages [44,45]. In fact, in our study, the most frequently symptoms reported was injection site pain, while the most frequently reported systemic events were fatigue and headache and systemic reactions (fever was more frequently reported after dose 2). All adverse events reported were mostly mild with resolution in one to two days. Furthermore, no differences in reactogenicity were noted between participants with previous SARS-CoV-2 infection. Our data are also in line with other recent reports conducted since the introduction of the vaccination in the 5–11-years age group [46,47,48]. In particular, Hause et al. [49] have analyzed data from three United State vaccine safety surveillance systems in the first four months after COVID-19 vaccine recommendations in this age group. In this report, among the 48,795 children enrolled, most reported reactions were mild-to-moderate, most frequently described the day after vaccination, and were more common after dose 2. The most common symptoms described were injection site pain, fatigue, headache, fever, and myalgia. No serious adverse events, such as myocarditis, occurred in our study. Myocarditis is a rare, but serious adverse event reported after COVID-19 vaccination (especially in adolescent males and young adult men) [50]. However, for children 5 to 11-years-old this condition was rarely reported after vaccination and reporting rates were lower than for older children comparable to the risk rate of the general population [51,52]. Finally, we found that patients with comorbidities had more adverse events than healthy patients. However, the reactions described were always mild, not interfering with normal daily activities, and no serious adverse events were reported. Therefore, we believe that vaccination should be promoted especially in these patients [53,54], who could be at higher risk of severe forms of COVID-19 [55,56].

### 4.2. Viral Variants and Vaccine Efficacy in Pediatric Age

Another concern about the pediatric COVID-19 vaccine is related to its efficacy against new SARS-CoV-2 variants. Data regarding this topic are still few and conflicting. Recently, Sacco and colleagues [57] described the effectiveness of BNT162b2 in children aged 5–11 years after the onset of the omicron variant (B.1.1.529) in Italy. The authors reported an adjusted vaccine effectiveness against infection of 29.4% in fully vaccinated and 27.4% in partially vaccinated children, with a decreasing trend at a distance after vaccination. Similar results have been shown in a study conducted on data from New York, in which vaccine effectiveness against infection in children aged 5–11 years decreased from 65% during the first 2 weeks after two doses to 12% after 28–34 days [58]. In a recent study, Chen et al. have studied the antibody response against the omicron variant in vaccinated children and children with SARS-CoV-2 infection, showing that only 38.2% of BNT162b2 vaccine recipients and 26.7% of recovered COVID-19 patients had neutralizing antibody serum titers, concluding finally that the pediatric age group will likely be more susceptible to vaccine breakthrough infections or reinfections due to the omicron variant than previous variants [59]. On the other hand, however, data from samples of adults are more encouraging. Lauring and others authors, in a study conducted on 11,690 patients, showed that mRNA vaccines were found to be highly effective in preventing COVID-19 associated hospital admissions related to the alpha, delta, and omicron variants. In particular, it was necessary to receive three doses of vaccines to achieve the same protection against omicron variant compared with two doses needed for the other variants [60]. Furthermore, in a recent study published in New England [61], the authors described the data based on 1185 children enrolled, showing that although two doses of vaccine results in lower protection against the omicron variant than the delta variant. However, this reduces the risk of critical illness caused by both variants also in pediatric age. Therefore, we believe that in view of the good safety profile of the vaccine in the age group of 5–11 years and the positive impact on hospitalizations and risk of critical illness (even during the period of the omicron variant), vaccination should continue to be promoted in this age group. Certainly, further studies are essential to confirm these data and especially to assess whether more booster doses are needed in the pediatric age group to achieve more effective and long-lasting protection [62].

### 4.3. Benefits of COVID-19 Vaccination in the Pediatric Age Group

We believe that the vaccination of BNT162b2 in children 5–11 years old is important for several reasons. Primarily, on the basis of mathematical models it has been demonstrated that vaccination against COVID-19 in children aged 5–11 years old leads to a significant reduction in the number of infections and, to a lesser extent, in the number of hospitalizations and deaths, with a reduction in the spread of coronavirus by up to 16% [63]. In fact, in our study, we demonstrated early protection already after a single dose and the absence of infection with a two-dose regimen in vaccinated children compared with unvaccinated children of the same age group. Vaccinating children will bring us closer to mass vaccination and consequently to the achievement of herd immunity, protecting in this way the most vulnerable population groups. In fact, it has been demonstrated that children’s transmission to household members is high and is similar to that from adults [64,65,66]. Second, despite in the pediatric age COVID-19 morbidity and mortality are significantly lower than in adults [67,68], children can also be at risk of developing post-COVID-19 complications, such as long COVID syndrome and multisystem inflammatory syndrome (MIS-C) [69,70,71]. Related to this, different reports in adults showed a decrease in disease complication rate COVID-19 in vaccinated people supporting the importance of the vaccination also in this age group [72]. Furthermore, MIS-C also appears to be much rarer in children and adolescents fully vaccinated [73,74]. Last but not least, children represent the category more susceptible to the indirect effects of the pandemic [75]. In fact, the closure of the school and of other extracurricular activities has caused an important psychosocial harm to children and especially adolescents during these years [76,77]. Therefore, the implementation of vaccines in this age group could enable children to re-engage in their world with positive effects on their psychophysical and relational development. But despite the fact that the beneficial effects of prevention of SARS-CoV-2 infection to seem to outweigh the potential risks of the vaccination in pediatric ages as well, there is still a low intention by parents to vaccinate their children (mainly due to fear of the side effects of vaccination) [78].

### 4.4. Limitations of the Study

This report is subject to some limitations (primarily the low sample size and the short surveillance period). Second, we also acknowledge that symptoms were only reported through an online survey rather than directly ascertained and are related to a single-center of study. Finally, we did not compare the differences in clinical outcomes of vaccinated and unvaccinated patients who contracted SARS-CoV-2 infection.

### 4.5. Conclusions

Pediatric vaccination with COVID-19 is recent, and vaccine hesitancy among parents of children aged 5–11 years is still high (despite the increase of COVID-19 cases even in this age group in recent months). Our findings support the data available in the literature on COVID-19 vaccination in children, showing a good safety profile of m-RNA vaccine in healthy children and those with comorbidities. We believe our data could be useful to reassure parents and promote pediatric vaccination. Certainly, further studies are needed to evaluate the efficacy of COVID-19 vaccine in children aged 5–11 years and the degree of protection against new variants

## Figures and Tables

**Figure 1 vaccines-10-01224-f001:**
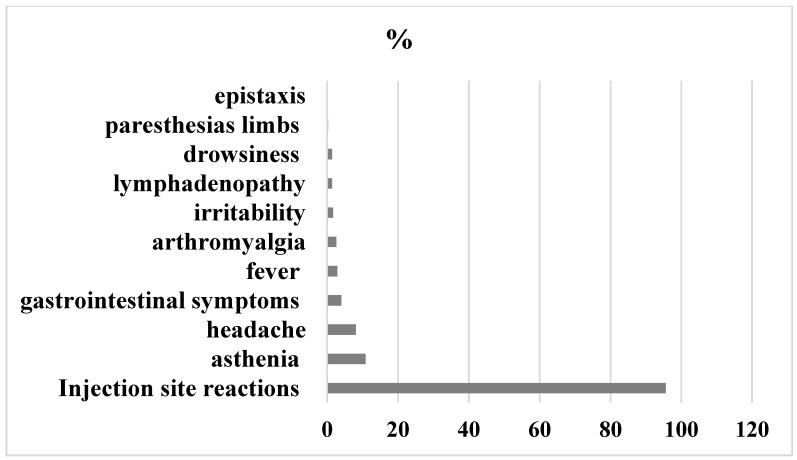
Adverse Events 24–48 h after first dose.

**Figure 2 vaccines-10-01224-f002:**
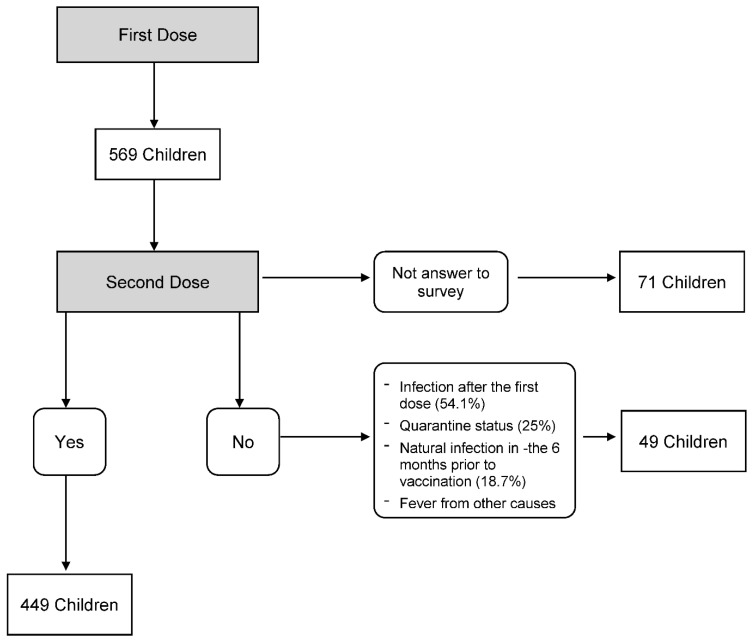
Flowchart patients enrolled.

**Figure 3 vaccines-10-01224-f003:**
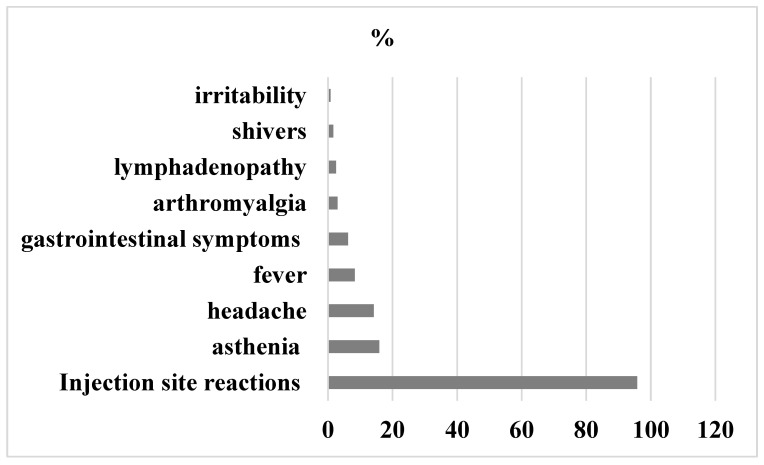
Adverse events 24–48 h after second dose.

**Table 1 vaccines-10-01224-t001:** Children’s clinical and demographic characteristics.

Children’s Characteristics		
Sex (F/M), % (n)	55.9 (318)/44.1 (251)	
Age (median ± DS), months	114 ± 4.24	
Comorbidities % (n)	None:	
	Comorbidity	80.7 (459)
	Allergies	19.3 (110)
	Asthma	46 (51)
	Gastrointestinal diseases	24 (26)
	Cardiac disease	9 (10)
	Neurological diseases	8 (9)
	Nephrological diseases	7 (8)
	Rheumatological diseases	3 (3)
	Other diseases	1 (1)
Previous SARS-CoV-2 infection % (n)	6.4 (37)	
Clinical course of the infection	Asymptomatic	41.7 (15)
	Symptomatic	58.3 (22)
	Fever	71.4 (16)
	Respiratory symptoms	42.8 (9)
	Headache	33.4 (7)
	Gastrointestinal symptoms	14.2 (3)
	Asthenia	9.5 (2)
	Arthralgias	4.8 (1)

## Data Availability

Not applicable.

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
