# Peer review of "Cross-Sectional Survey on BNT162b2 mRNA COVID-19 Vaccine Serious Adverse Events in Children 5 to 11 Years of Age: A Monocentric Experience"

_vaccines, 2022, doi:10.3390/vaccines10081224_

Round 1
Reviewer 1 Report
Thank you for your submission on this timely topic. I have a few comments/suggestions
Please define any abbreviations before use (ASL etc.)
Methods:
Was the survey secured or password protected? How was this health information kept secured?
Was there ever written consent- how could you verify whom you were speaking with if just had a verbal consent?
I am not clear if it was the doctors were answering the survey questions or the parents of the minors. Did the parents give consent to release health information about their children?
There are a number of formatting/grammar issues to review: example: page 4 line 135 has a closed parenthesis but no open parenthesis. A number of sentences are started with a number, consider changing this format. Figure 2 should that read "No answer to survey"?
Page 6 line 192 consider adding (acetaminophen) next to paracetamol
Discussion: consider making paragraphs for each new topic not one paragraph
References:
#1 was January 17 date of access? please add accessed Jan 17, 2022
#9 what was the date this website was accessed?
#10 what was the date this website was accessed?
#12 what was the date this website was accessed?
#18 please correct has the date 3/3/2021 at the start not the author's name
#22 please correct has the "a" that needs to be removed
#26, 27, 28, 29 Are there a doi numbers for these references?
#34 Is there a doi for this reference?
#35 when was this website accessed?
#38 similar to #22 has an "a" that must be deleted
Author Response
Dear Editors and Reviewers,
Thank you for your consideration of our manuscript. We truly think the manuscript is improved after the revisions suggested. Below we respond in detail to the comments and points the reviewer raised. We now submit our revised manuscript for publication in Vaccines. We marked the revisions through the manuscript using the “track changes”.
Reviewer 1
Thank you for your submission on this timely topic. I have a few comments/suggestions
Please define any abbreviations before use (ASL etc.)
We have now defined all abbreviations in the text.
Methods:
Was the survey secured or password protected? How was this health information kept secured?
Yes, the survey was protected by password. The web link produced to collect the data was accessible through a password only to pediatricians involved in the survey.
Was there ever written consent- how could you verify whom you were speaking with if just had a verbal consent?
At the time of vaccination, parents gave written consent to receive the vaccine and were informed that they would be contacted by a doctor in our department to check for any side effects after vaccination. Children vaccinated in the Latina ASL are registered through a database that collects the child's biographical information, date of vaccination, parents' first and last names, and telephone contact information. Through this database, we contacted the parents of children who were vaccinated during the study period and asked them for their verbal consent to participate in the research before starting the survey.
I am not clear if it was the doctors were answering the survey questions or the parents of the minors. Did the parents give consent to release health information about their children?
Parents of minors responded to the survey questions. Before starting the survey, we asked them for their verbal consent to participate in the research and release health information about their children. The initiative of this research was welcomed with great enthusiasm by parents given the recent introduction of the covid-19 vaccine in pediatric age.
There are a number of formatting/grammar issues to review: example: page 4 line 135 has a closed parenthesis but no open parenthesis. A number of sentences are started with a number, consider changing this format. Figure 2 should that read "No answer to survey"?
We have now corrected formatting/grammar issues through the whole text; we have now deleted the parenthesis on page 4 line 135 and modified figure 2.
Page 6 line 192 consider adding (acetaminophen) next to paracetamol
We have now added acetaminophen next to paracetamol
Discussion: consider making paragraphs for each new topic not one paragraph
We have now modified the discussion making different paragraphs related to different topics; in particular we have divided the discussion in different sections: Safety of COVID-19 vaccine in pediatric age, efficacy of covid-19 vaccine against new variants; benefits of covid-19 vaccination in pediatric age and the limitation of the study.
References:
#1 was January 17 date of access? please add accessed Jan 17, 2022
Yes, we have now added data of access.
#9 what was the date this website was accessed?
The date of access was 29 October 2021. We have now corrected this reference.
#10 what was the date this website was accessed?
25 November 2021, we have now added this information in the reference
#12 what was the date this website was accessed?
10 December 2021, we have now added this information in the reference.
#18 please correct has the date 3/3/2021 at the start not the author's name
Done
#22 please correct has the "a" that needs to be removed
Done
#26, 27, 28, 29 Are there a doi numbers for these references?
We have now added the doi numbers for all references
#34 Is there a doi for this reference?
We have now added the doi number
#35 when was this website accessed?
23 June 2021, we have now specified the date of access
#38 similar to #22 has an "a" that must be deleted
Done
Reviewer 2 Report
Please improve introduction with relevant references, Sample size is low please justify. Result and discussion can be improved with discussing the result with appropriate references. Novelty of the study is missing which can be included in the separate part as conclusion.
Author Response
Dear Editors and Reviewers,
Thank you for your consideration of our manuscript. We truly think the manuscript is improved after the revisions suggested. Below we respond in detail to the comments and points the reviewer raised. We now submit our revised manuscript for publication in Vaccines. We marked the revisions through the manuscript using the “track changes”.
Reviewer 2
Please improve introduction with relevant references.
We have now improved and updated the introduction with more references.
In particular, we updated new covid-19 cases and considered the increase in recent months related to new variants. We included all vaccines approved since 2021 according to phase 3 studies and included many reports conducted in different countries related to the positive impact of vaccination on the incidence of infection, hospitalizations, and deaths. Finally, we have also updated the section on approved vaccines in pediatric age. We thank the reviewer, in fact we believe that in this way the introduction is improved and more complete since the research on covid-19 is constantly increasing.
Sample size is low please justify.
We agree with reviewer. We have now added the low sample size as limitation of the study.
Result and discussion can be improved with discussing the result with appropriate references.
We have now improved the discussion adding appropriate and updated references. In In particular, we divided the discussion into paragraphs. The first part regarding the safety of the covid-19 vaccine in the 5-11 age group, comparing our results with those of the currently available and more recent studies; the second part regarding the efficacy of the covid-19 vaccine against the new variants, especially the omicron variant responsible for this latest wave; in the last part we have included our considerations on the positive effects of vaccination even in this age group and the limitations of the study
Novelty of the study is missing which can be included in the separate part as conclusion.
We have now added the section of conclusion:
“Pediatric vaccination with Covid-19 is recent, and vaccine hesitancy among parents of children aged 5-11 years is still high, despite the increase of covid-19 cases even in this age group in recent months. Our findings support the data available in the literature on covid-19 vaccine in children, showing a good safety profile of m-RNA vaccine in healthy children and those with comorbidities. We believe our data could be useful to reassure parents and promote pediatric vaccination. Certainly, further studies are needed to evaluate the efficacy of covid-19 vaccine in children aged 5-11 years and the degree of protection against new variants”
Reviewer 3 Report
This study is based on a survey of 569 children in Italy, whose adverse events were followed after 1st and 2nd vaccination with BNT162b2 mRNA COVID-19 vaccine.
The main issues with the study are the low number of subjects to detect any rare and serious adverse events and the short follow-up time. However, it is also important to have real-life data on the vaccines in children. The authors should bring the problems of the study design (that they already do) more up in discussion.
Furthermore, some references and discussion on the new virus variants is needed since the protection against covid infection has also been reduced, which also reduces the utility of current covid vaccines in children.
Author Response
Dear Editors and Reviewers,
Thank you for your consideration of our manuscript. We truly think the manuscript is improved after the revisions suggested. Below we respond in detail to the comments and points the reviewer raised. We now submit our revised manuscript for publication in Vaccines. We marked the revisions through the manuscript using the “track changes”.
Reviewer 3
This study is based on a survey of 569 children in Italy, whose adverse events were followed after 1st and 2nd vaccination with BNT162b2 mRNA COVID-19 vaccine.
The main issues with the study are the low number of subjects to detect any rare and serious adverse events and the short follow-up time. However, it is also important to have real-life data on the vaccines in children. The authors should bring the problems of the study design (that they already do) more up in discussion.
We agree with reviewer; we have now added a new paragraph “Limitations of the study” and we have described all limitations of the study: the low sample size, the short follow up, a single-centre of study and the fact that the symptoms were only reported through online survey.
Furthermore, some references and discussion on the new virus variants is needed since the protection against covid infection has also been reduced, which also reduces the utility of current covid vaccines in children.
We have now added a new section in the discussion “Viral variants and vaccine efficacy in pediatric age” with the most recent evidence on this topic:
Another concern about the pediatric covid-19 vaccine is related to its efficacy against new SARS-CoV-2 variants. Data regarding this topic are still few and conflicting. Recenly, Sacco and colleagues described the effectiveness of BNT162b2 in children aged 5–11 years after the onset of omicron variant (B.1.1.529) in Italy. The authors reported an adjusted vaccine effectiveness against infection of 29·4% (95% CI 28·5–30·2) in fully vaccinated and 27·4% in partially vaccinated children, with a decreasing trend at a distance after vaccination. Similar results ere shown in a study conducted on data from New York (NY, USA), in which vaccine effectiveness against infection in children aged 5–11 years decreased from 65% during the first 2 weeks after two doses to 12% 28–34 days. On the other hand, however, data from samples of adults are more encouraging. Lauring and others authors, in a study conducted on 11 690 patients, showed that mRNA vaccines were found to be highly effective in preventing covid-19 associated hospital admissions related to the alpha, delta, and omicron variants; in particular it was necessary three doses of vaccines to achieve the same protection against omicron variant compared with two doses needed for the other variants Furthermore, in a recent study published on New England, the authors described the data based on 1185 children enrolled, showing that although two doses of vaccine results in lower protection against the omicron variant than the delta variant, however, it reduces the risk of critical illness caused by both variants also in pediatric age. Therefore, we believe that in view of the good safety profile of the vaccine in the age group of 5-11 years and the positive impact on hospitalizations and risk of critical illness even during the period of the omicron variant, vaccination should continue to be promoted in this age group. Certainly, further studies are essential to confirm these data and especially to assess whether more booster doses are needed in the pediatric age group to achieve more effective and long-lasting protection. (References: Sacco et al. Effectiveness of BNT162b2 vaccine against SARS-CoV-2 infection and severe COVID-19 in children aged 5–11 years in Italy: a retrospective analysis of January–April, 2022. Lancet 2022; Dorabawila V et al. Effectiveness of the BNT162b2 vaccine among children 5–11 and 12–17 years in New York after the emergence of the omicron variant. medRxiv 2022; Chen LL. et al. Omicron variant susceptibility to neutralizing antibodies induced in children by natural SARS-CoV-2 infection or COVID-19 vaccine. Emerg Microbes Infect. 2022; Lauring AS et al. Clinical severity of, and effectiveness of mRNA vaccines against, covid-19 from omicron, delta, and alpha SARS-CoV-2 variants in the United States: prospective observational study. BMJ. 2022.; Price AM et al. BNT162b2 Protection against the Omicron Variant in Children and Adolescents. N Engl J Med. 2022)
Reviewer 4 Report
The study title should at least read as: Cross-sectional survey on BNT162b2 mRNA COVID-19 vaccine serious adverse events in children 5 to 11 Years of Age: A monocentric experience
In the Abstract: The incidence of SARS-CoV-2 infection was higher in unvaccinated subjects than in those who received the first dose (p < 0.0001); after the second dose, none of vaccinated contracted the infection in our period of study.
The BNT162b2 mRNA COVID-19 vaccine does not protect SARS-CoV-2 infection, but reduce severity of disease and hospitalization as such this point should not be included
The tertiary aim of the study was not thoroughly analyzed and compared, here are some questions:
The tertiary aim of the study was not thoroughly analyzed and compared, here are some questions: Where the children infected at a higher risk or not (i.e. Establish the risk levels of children both infected and uninfected: Parent/s and sibling exposed the child, school or daycare attendance). The analysis and methodology to compare the two groups is not well indicated. The vaccine does not protect infection but assists in severity of disease (i.e. hospitalization and ending up in ICU, severe forms of COVID-19), and such the study should also have compared how many ended in hospital with severe COVID (or required assistant to breathing)
Author Response
Dear Editors and Reviewers,
Thank you for your consideration of our manuscript. We truly think the manuscript is improved after the revisions suggested. Below we respond in detail to the comments and points the reviewer raised. We now submit our revised manuscript for publication in Vaccines. We marked the revisions through the manuscript using the “track changes”.
Reviewer 4
The study title should at least read as: Cross-sectional survey on BNT162b2 mRNA COVID-19 vaccine serious adverse events in children 5 to 11 Years of Age: A monocentric experience
We have now modified the title, as suggested by reviewer.
In the Abstract: The incidence of SARS-CoV-2 infection was higher in unvaccinated subjects than in those who received the first dose (p < 0.0001); after the second dose, none of vaccinated contracted the infection in our period of study.The BNT162b2 mRNA COVID-19 vaccine does not protect SARS-CoV-2 infection, but reduce severity of disease and hospitalization as such this point should not be include
We agree with reviewer that the BNT162b2 mRNA COVID-19 vaccine mainly reduces severity of disease and hospitalization in both pediatric and adult patients (Tartof SY, et al. Effectiveness of mRNA BNT162b2 COVID-19 vaccine up to 6 months in a large integrated health system in the USA: a retrospective cohort study. Lancet 2021; Reis BY, et al. Effectiveness of BNT162b2 vaccine against Delta variant in adolescents. N Engl J Med 2021;385: 2101–3; Olson SM et al.; Overcoming Covid-19 Investigators. Effectiveness of BNT162b2 vaccine against critical Covid-19 in adolescents. N Engl J Med 2022;386 :713–23; Lutrick K, et al. Interim estimate of vaccine effectiveness of BNT162b2 (Pfizer-BioNTech) vaccine in preventing SARS-CoV-2 infection among adolescents aged 12–17 years—Arizona, July–December 2021. MMWR Morb Mortal Wkly Rep 2021; 70:1761). However, m-RNA vaccines have also been shown to be effective in preventing SARS-CoV-2 infection (Thompson MG, et al. Prevention and Attenuation of Covid-19 with the BNT162b2 and mRNA-1273 Vaccines. N Engl J Med. 2021). Furthermore, recently it was been demonstrated that m-RNA vaccines attenuated the viral RNA load in breakthrough infection resulting in lower infectiousness and contributing to vaccine effect on virus spread (Levine-Tiefenbrun M, et al. Initial report of decreased SARS-CoV-2 viral load after inoculation with the BNT162b2 vaccine. Nat Med. 2021 May;27(5):790-792).
Therefore, we have now deleted this point in the abstract, since our study is mainly based on vaccine safety in children aged 5-11 years, and we have kept the description of this difference in the incidence of infection over the study period between vaccinated and unvaccinated subjects only in the results section.
The tertiary aim of the study was not thoroughly analyzed and compared, here are some questions: Where the children infected at a higher risk or not (i.e. Establish the risk levels of children both infected and uninfected: Parent/s and sibling exposed the child, school or daycare attendance). The analysis and methodology to compare the two groups is not well indicated. The vaccine does not protect infection but assists in severity of disease (i.e. hospitalization and ending up in ICU, severe forms of COVID-19), and such the study should also have compared how many ended in hospital with severe COVID (or required assistant to breathing).
For the tertiary aim of the study, we have considered all children resident in the ASL of Latina and we have analyzed two time periods: the first from seven days after the first dose (December 22) to January 4; the second from seven days after the second dose (January 12) to January 31. Then, we compared the number of children infected in the unvaccinated group and in the vaccinated group in the two time periods of study through the chi square test.
We agree with reviewer, we are not aware of the risk factors of the infected children and the clinical outcomes (hospitalizations and severe form of covid-19) of both vaccinated and unvaccinated children who became infected, so we added these considerations into the limitations of the study.
Round 2
Reviewer 4 Report
The manuscript has been sufficiently improved to warrant publication in Vaccines